# Understanding Ocular Findings and Manifestations of Systemic Lupus Erythematosus: Update Review of the Literature

**DOI:** 10.3390/ijms232012264

**Published:** 2022-10-14

**Authors:** Wojciech Luboń, Małgorzata Luboń, Przemysław Kotyla, Ewa Mrukwa-Kominek

**Affiliations:** 1Department of Ophthalmology, Faculty of Medical Sciences, Medical University of Silesia, 40-514 Katowice, Poland; 2Department of Ophthalmology, Professor K. Gibiński University Clinical Center of the Medical University of Silesia, 40-055 Katowice, Poland; 3Department of Internal Medicine, Rheumatology and Clinical Immunology, Medical University of Silesia, 40-055 Katowice, Poland

**Keywords:** systemic lupus erythematosus, autoimmune, ocular complications

## Abstract

Systemic lupus erythematosus (SLE) is a chronic multisystem autoimmune disease. Up to one-third of patients suffering from SLE have various ocular manifestations. The ocular findings may represent the initial manifestation of the systemic disease and may lead to severe ocular complications, and even loss of vision. Ocular manifestations are often associated with degree of systemic inflammation, but also can precede the occurrence of systemic symptoms. Early diagnosis and adequate management of patients with SLE are crucial and require cooperation between various specialists. Proper preparation of ophthalmologists can help to differentiate between complication of SLE and other ocular disorders. New therapies for SLE are promising for potential benefits, however, ocular side effects are still unknown.

## 1. Introduction

### Systemic Lupus Erythematosus

Systemic lupus erythematosus (SLE) is a chronic autoimmune disease of the connective tissue that may have deleterious effects on many systems and organs of the body, including eyes, often with relapsing clinical course. The clinical picture of SLE is very diversified and depends on the affected systems and organs, therefore, it should be considered as a systemic disease and not as a specified one to a certain organ.

Systemic lupus erythematosus was first described and distinguished from the other diseases manifesting erythema in 1833, and the first biopsy of skin lesions was performed in 1872. In 1929, the first description of the presence of ocular complications of SLE appeared. 

In 1933, Semon and Wolff described the histopathological features of choroiditis and subretinal fluid in the course of SLE [1].

It is estimated that as many as one-third of SLE patients experience ophthalmic symptoms ranging from relatively mild to severe, threatening eyesight. The involvement of the eye structures depends on the systemic activity of the disease and in many cases precedes the involvement of other vital organs (e.g., kidneys, lungs, heart). Ophthalmologists who care for patients suffering from SLE should pay special attention to the periods of exacerbation of the disease and the possible side effects of immunosuppressive drugs [2].

## 2. Ophthalmic Manifestations of SLE

Ophthalmic manifestations depend on the susceptibility of the patient and may correlate with the systemic activity of the underlying disease. Ophthalmic changes may occur at the beginning of the disease or appear during its development. Ocular involvement is described as low to moderately common in the course of SLE, however there have been reported cases indicating that it may be a serious vision hazard. Disease manifestations may include abnormalities of the protective apparatus of the eye, mainly the eyelids, the area of the orbit, the appendages of the eye, all structures of the eye itself and the optic nerve. The most common diagnosed conditions during SLE are keratoconjunctivitis sicca (dry eye disease), iritis and ciliary body inflammation and retinal vascular changes, while optic neuritis and occlusive vasculitis are the most damaging to vision and cause blindness. Active inflammation located in the retina and choroid can mimic general vasculitis that can occur in other organs [3]. Pathological changes located in the posterior segment of the eye (optic nerve, retina, uvea) most often precede other systemic features of the disease and may help in the diagnosis and adequate implementation of therapy for SLE [4].

Ophthalmologists should be involved in the examination and follow-up of patients in cooperation with trained specialists of other professional profiles, usually rheumatologists from the beginning of the diagnosis of SLE, even in the case where the disease remains clinically silent and patients do not show ophthalmic symptoms. Physicians undertaking the examination of patients with autoimmune diseases should perform basic ophthalmic examinations: distant and near visual acuity, color perception, visual field examination, intraocular pressure measurement, Amsler grid, thorough examination of the anterior segment of the eye with a slit lamp, indirect and direct ophthalmoscopy of the posterior segment of the eye, as well as additional specialist examinations: mfERG, flash-VEP, BM, OCT, FAF, FFA, and ICGA [5,6].

### 2.1. Orbit

A rare presentation of SLE is involvement of the orbital tissues. Subcutaneous orbital tissue inflammation secondary to SLE was first described by Kaposi in 1883 [1]. The clinical picture described in the literature may take the form of orbital inflammation, infarction of orbit muscles or myositis, panniculitis, manifestation of exophthalmos and orbital edema. Patients’ symptoms and signs include orbital pain, blurred vision, exophthalmos, eyelid edema, chemosis, contracture, and reduced mobility of the extraocular muscles [7]. The described cases include both bilateral and unilateral orbital involvement, and inflammatory masses in the orbit, often undiagnosed until complications arise, may induce symptoms of pseudotumor [8]. Vasculitis in the orbit leads to a lack of perfusion of the eyeball and extraocular muscles, which may result in reduced mobility of the eyeball and cause irreversible loss of vision due to ischemic damage to the optic nerve. The consequence of orbital vasculitis may also be excessively elevated intraocular pressure in the course of neovascular glaucoma [9].

Orbital computed tomography or ultrasound showing enlargement of one or more extraocular muscles may be in combination with the patient’s symptoms mentioned above. In laboratory tests, levels of creatine kinase, aldolase, CRP, procalcitonin and myoglobins are often markedly elevated. Tissue biopsy (histopathology indicates perivascular lymphocytic infiltration), serological examination, long-term follow-up and systemic evaluation may be necessary to facilitate proper diagnosis and exclusion of other possible causes such as thyroid orbitopathy, other types of inflammation, bacterial infections (most often induced by staphylococcus epidermidis or aureus in this area) and also neoplastic processes. Orbital tissue inflammation in SLE can be limited to the orbit itself or spread to adjacent tissues, which can result in blindness due to optic neuropathy [10]. Chan et al. recently described a case of a 45-year-old woman with orbital myositis in the course of SLE, where therapy with prednisolone was successful, and switching from azathioprine to mycophenolate mofetil. The improvement of the local condition was thoroughly proven in CT-scans [11].

Treatment of orbital inflammation is carried out with systemic immunosuppressive drugs and usually responds to therapeutic doses of steroids [9,10].

Orbital inflammation is often associated with discoid lupus erythematosus. Clinical symptoms include tender, deep subcutaneous nodules. Discoid lupus erythematosus (DLE) is chronic lupus erythematosus with only skin manifestation without involvement of internal organs. Typical changes in DLE are slightly raised, scaly, and atrophic. They are most common on the head, face (including the orbital area), neck and other areas exposed to sunlight, especially in young and middle-aged women [12].

Inflammation of the block apparatus of the superior oblique muscle (trochleitis) constitutes one of the infrequently detected symptoms of orbital lupus, but can also be a manifestation of SLE. The published clinical case describes sequential bilateral trochleitis as a feature presented in the course of SLE [13].

### 2.2. Periorbital Tissues

Periorbital inflammation and edema are rare symptoms of systemic and discoid lupus erythematosus. In the course of DLE, due to the increased permeability of blood vessels and the accumulation of mucin deposits in the subcutaneous tissue, develop chronic dermatitis without necrosis [14].

The differential diagnosis should initially include basal cell carcinoma, squamous cell carcinoma, Bowen’s disease, actinic keratosis, contact dermatitis, alopecia areata, atopic dermatitis, seborrheic dermatitis, psoriasis, pemphigus, albinism and sarcoidosis. Treatment options include observation, topical and systemic corticosteroids (mainly methylprednisolone), and antimalarial drugs (e.g., hydroxychloroquine). Periorbital heliotropic edema is a characteristic cutaneous symptom of dermatomyositis, rarely but also common in systemic lupus erythematosus. Approximately 5% of SLE cases show this type of edema [15,16]. In case of clinical suspicion, rapid histopathological examination is of key importance for early diagnosis.

Lo et al. described a clinical case of a 12-year-old female without registry of formerly diagnosed systemic diseases who had persistent heliotropic erythema around the orbit and subcutaneous tissue edema, which was the first symptom of SLE [15].

Mullaaziz et al. recently described a recent case of a 43-year-old woman with bilateral lupus edema and subcutaneous involvement of the periorbital area. Treatment with hydroxychloroquine and pimecrolimus for up to 6 months was shown to be effective [17].

### 2.3. Eyelids

The skin of the eyelids and around the eyes may be involved in SLE in a different way than in the classically described cheek rash. It may include the presence of scaly, pigmented lesions on the eyelids that are present in both SLE and DLE. Although the course of DLE on the eyelids is usually mild, diagnosis is often difficult, resulting in patients being given different recommendations and, consequently, negative treatment outcomes. Eyelid involvement has also been reported in cutaneous lupus erythematosus. The most common reasons for reporting patients to an ophthalmologist are chronic erythema of the eyelids and meibomian gland inflammation. Often, unilateral or bilateral blepharitis coexists. Clinically, the lesions of the eyelids appear as lumps or hardened plaques that often adhere to the overlying skin. Lupus discoid rash on the eyelids usually appears on the lower eyelid as an irritating, discreet, slightly raised erythematous scaly plaque that may encompass the eyelid margin [18]. Lupus involvement of the eyelid margins can impose a direct and significant impact on a troublesome and difficult to treat condition—madarosis—in which excessive eyelash loss leads to the weakening of the protective barrier of the eyeball. Once properly treated and effectively healed, the lesions of the eyelids may fade away leaving residual scars, or even lead to ectropion or entropion scarring. These changes of the eyelid margins very often require surgical intervention and appropriate oculoplastic treatment in order to restore the efficiency and functionality of the protective apparatus of the eye. Misdiagnosis can lead to eyelid margin deformation and delay diagnosis of systemic lupus [19,20].

In order to confirm the diagnosis, eyelid skin biopsies should be performed followed by immunohistochemical examination. It is common that in the collected biopsy, perivascular infiltrations with lymphocytes and degeneration of the basal layer are detected [3,16]. 

Cakci et al. described a 40-year-old female continuously suffering a squamous, erythematous, slightly swollen and atrophic skin area with scaly patches and telangiectasia on the eyelids, and performed a biopsy of the right upper eyelid based on suspected diagnosis of DLE. Histological examination revealed epidermal atrophy, vacuolar degeneration and lymphocytic infiltrates in deep and superficial perivascular and appendage areas, confirming the diagnosis of DLE [14].

Nagendram et al. described the case of a 32-year-old woman who developed characteristic skin patches within 6 months and loss of eyelashes on the lower eyelid—these symptoms led to the diagnosis of DLE. Lesions were successfully treated with prednisolone and hydroxychloroquine in combination with topical mometasone ointment [21].

Treatment of eyelid lesions in SLE and DLE is usually performed with systemic anti-inflammatory drugs, topical corticosteroids and oral antimalarial drugs [20].

### 2.4. Surface of the Eye

The most frequently described ophthalmic symptom in the course of SLE is dry keratoconjunctivitis, also known as dry eye disease (DED), affecting up to one third of patients (25–35%) [22].

Allergic disease of conjunctiva (including allergic conjunctivitis, atopic keratoconjunctivitis, vernal keratoconjunctivitis and giant papillary conjunctivitis) are rarely diagnosed in SLE patients. In the studies, people with SLE were shown to be at increased risk of IgE-related allergic disorders and atopic diseases in comparison with control groups [23].

In their extensive meta-analysis from 2021, Wang et al. showed that more than 25% of patients with SLE present abnormalities in the ophthalmic Schirmer test, and 12% of patients meet the criteria for secondary Sjogren’s syndrome [24].

### 2.5. Episcleral and Sclera

Episcleritis in the course of SLE is typically seen in young women with relatively mild course, and the symptoms include tearing, dull pain in the eyeball and redness of the eye surface due to dilation of the superficial blood vessels (Figure 1), which shrink significantly when phenylephrine is applied topically. Deterioration of visual acuity and severe pain in the eyeballs are rare. Episcleritis and scleritis have been reported to be associated with several systemic autoimmune diseases. The development of episcleritis in patients with SLE is attributed to the deposition of immunoglobulins in the tissues [2,25].

Read et al. found that six out of nine patients in the described series of pediatric episcleritis had systemic connective tissue disease, including SLE [26]. Although relatively rare, episcleritis and scleritis may be initial symptoms and precede other manifestations of SLE, and their occurrence is usually indicative of ongoing, active disease. Episcleritis is usually self-limiting disease that does not require treatment. In severe cases, topical non-steroidal or steroid eye drops and vasoconstrictor drops may be required. It is rarely required to use systemic or periocular injection of corticosteroids. A solution, in the case of failure to achieve the therapeutic goal after the use of topical drugs, may be oral administration of NSAIDS in high doses for a period of several days [27].

Scleritis according to its features of occurrence can be classified as anterior or posterior. Anterior scleritis in patients with SLE may present as diffuse or nodular inflammation, with symptoms such as redness, soreness, tenderness and tearing. The nodular type of inflammation more often leads to tissue necrosis. Depending on the pathological features, necrotizing anterior scleritis may be defined as vascular obstruction, granulation or scleromalacia. The injected deep episcleral vessels produce a purple discoloration of the sclera tissue, best visible in natural light. The patient undergoing the treatment performed in such manner may experience periodic pain in the face, often extending to the eye area, during rest. Inflammation is more likely to occur unilaterally. Posterior scleritis is not necrotic and is rare in SLE patients. It is not associated with “red eye” because it affects the sclera located posterior to the equator of the eyeball. Symptoms include eye pain, blurred vision, limited eye movement, double vision (diplopia), and exophthalmos. Blurred vision is most often caused by exudative detachment of the retina, deformation of the macular area due to a large mass of the sclera, and cystic macular edema. Scleritis can also lead to mild uveitis [25]. Scleritis may indicate activity of an underlying systemic disease that requires systemic treatment (including steroids and immunosuppressants) [28]. Changes in the sclera can be visualized generally by OCT, however type B ultrasound of the eyeball, computed tomography (CT) and magnetic resonance imaging (MR) have also been shown to be diagnostic. Scleral signal enhancement, focal cellulitis around the sclera and thickening of the sclera wall, but without perforation, are usually detected by MR in the active phase [29]. Treatment of scleritis includes topical steroids and NSAIDs in the form of drops, and in addition, systemic drugs—steroids, hydroxychloroquine, exceptionally cyclophosphamide or leflunomide [30].

### 2.6. Cornea

Corneal epitheliopathy, scarring, peripheral ulcerative keratitis, filamentous keratitis, interstitial keratitis and endotheliitis with corneal edema may occur secondary to keratoconjunctivitis sicca in SLE (Figure 2 and Figure 3). Corneal symptoms that may appear as the initial or later manifestations of SLE have various clinical features and may be a marker of active systemic inflammation. The patient often complains of eye pain, redness, excessive tearing, hypersensitivity to light, decreased visual acuity and blurred vision [31].

It has been proven that SLE patients present with thicker corneal periphery than controls characteristically sparing the superior quadrant. Possible corneal photosensitivity leading to peripheral immune complex deposition as well as flatter posterior corneal surface at the periphery are proposed explanations for these findings [32].

Peripheral ulcerative keratitis (PUK) is relatively rare in SLE, more often associated with rheumatoid arthritis, polyarteritis nodosa and inflammatory bowel diseases, however some cases of PUK can be sight-threatening. The appearance of PUK often indicates the presence of active vasculitis [33].

PUK occurs when immune complexes activate the complement system, causing chemotaxis of inflammatory cells (neutrophils and macrophages) and the release of collagenases and proteases, which lead to the destruction of the corneal stroma. Pro-inflammatory cytokines stimulate the stromal keratocytes to produce matrix metalloproteases that contribute to the breakdown of the cornea [34]. MMP-1 and MMP-8 attract inflammatory cells near the limbus, where the destructive process of the cornea begins [35]. PUK manifests as a devastating crescent-shaped lesion in the stroma of the cornea. Progressive thinning can lead to the formation of descemetocele and, consequently, perforation of the cornea. Patients report, on a regular basis, an experience of pain, redness, tearing, photophobia and decreased vision [36]. The primary goals of treatment of PUK are minimizing inflammation, preventing bacterial superinfections, promoting ulcer healing and preventing perforation. Indications for surgical intervention include perforation of the cornea or excessive thinning of the cornea with the risk of perforation. Surgical options include layered, penetrating or tectonic keratoplasty as well as optional corneo-scleral keratoplasty with partial sclera resection, in the case of sclera malacia [33,34].

In patients with corneal edema, endothelial cell dysfunction can be demonstrated by spectral microscopy. Immune complexes in SLE can build up in the corneal basement membrane and limbal vascular endothelial cells which can lead to the release of chemotactic cytokines and induce peripheral inflammatory infiltrates of the cornea. Corneal involvement in SLE may include the epithelium and present as superficial punctate keratitis. SLE-related corneal infiltrates should be treated with topical and systemic steroids supplemented with additional therapy to moisturize and regenerate the surface of the eye.

According to some authors, autoimmune diseases such as SLE are associated with keratoconus, which may indicate the role of the immune system in the pathogenesis of keratoconus [37]. On the other hand, there are publications which state that there is no significant association between keratoconus and SLE [2].

### 2.7. Retina

Lupus retinopathy occurs in approximately 10% of patients, although the incidence appears to decrease with improvement of systemic therapy [38].

In times when steroid therapy was not widely used, up to half of SLE patients showed signs of retinopathy during their lifetime. However, with the advent of steroids and immunosuppressive therapy, the incidence of retinopathy has significantly decreased. Mild retinopathy can be asymptomatic and incidental, while severe vascular occlusive retinopathy presents with visual deterioration, distortions and visual field defects. A total of 72% of eyes with SLE retinopathy were found to present retinal neovascularization with sequelae such as vitreous hemorrhage (63%) and retinal detachment (27%). Other retinal symptoms include micro aneurysms, narrowing of the arteries, lesions at arteriovenous junctions, retinal edema, retinal exudation and multifocal serous detachment of the retina or pigment epithelial detachment. SLE-related retinal microangiopathy is believed to result from immune complex-mediated vascular damage and the thrombosis of the retinal microcirculation. Histopathology shows deposits of immunoglobulins and complement components, perivascular unicellular infiltration and less frequently, fibrinoid necrosis. Antiphospholipid antibodies (anti-cardiolipin antibodies or lupus anticoagulant) and anti-beta-2 glycoprotein-I may play a key role in some patients in the pathogenesis of lupus retinopathy [39,40,41]. In 1984, Hall et al. for the first time reported an association between severe retinal lupus vasculopathy and the presence of antiphospholipid antibodies [42]. It is well known that recurrent infarcts and thromboembolic diseases are hallmarks of the antiphospholipid syndrome (APS). In a study by Montehermoso et al., antiphospholipid antibodies were detected in 77% of patients with lupus disease of the retina or optic nerve, compared with only 29% of patients with SLE without ocular involvement [43]. Studies using fluorescein angiography to visualize blood flow in the retina describe the hyperpermeability of arterioles and venules as well as the absence of capillary perfusion of the retinal vessels.

In the course of SLE, very common arterial hypertension secondary to autoimmune inflammatory kidney disease causes changes in the bloodstream to the fine arteries of the retina, causing their walls to stiffen and blood flow to be disrupted. Retinal vasculitis in SLE can result in severe obstruction and infarction of the relevant area of the eye’s retina [44].

Severe vascular occlusive retinopathy tends to be infrequent and, thus, a rarely signalized but well-described entity that is associated with an extensive lack of perfusion of the retinal capillaries, multi-branch obstruction of the central retinal artery, fundus neovascularization, vitreous hemorrhage, traction-related retinal detachment, and neovascular glaucoma resulting in significant loss of vision [45]. This most severe form of lupus retinopathy presents with a broad spectrum of ischemia, from obstruction in major vessels, such as the central retinal vessels and cilioretinal artery, to extensive microembolization in small vessels (Purtscher-type retinopathy).

A published report on Purtscher retinopathy in eight patients revealed an association between Purtscher retinopathy, lupus CNS involvement and high disease activity; the final improvement in visual acuity was usually low despite relatively quick treatment [46]. Other published scientific reports showed that 55% of the eyes of SLE patients with severe retinal vaso-occlusion showed loss of vision, often with visual acuity less than 20/200 [45,47].

The authors also found that CNS involvement by lupus was a common accompanying symptom in patients with such pronounced vascular changes in the retina. There have also been reports of pseudo-pigmentary retinitis with pigmentation disturbances, possibly secondary to vascular obstruction. Other reported changes in the retina of the eye in SLE include hard exudates and scarring. In the differential diagnosis of patients with SLE and retinitis, there should be consideration of severe uncontrolled hypertension, diabetes, Behçet’s disease, impending vena cava obstruction, sarcoidosis, syphilis, borreliosis, toxocarosis, HIV-associated retinopathy and cytomegalovirus retinitis, cancer-associated retinopathy (CAR) and melanoma-associated retinopathy (MAR) [48].

It has been assumed that the treatment of severe inflammatory retinal involvement in SLE is performed with intravenous corticosteroids, initially in the form of pulses. An alternative route of administration is the injection of corticosteroids in the area of the eyeball (periocularly), but this requires more caution and knowledge of the appropriate technique. It should be avoided in the case of excessive tissue swelling after injection and consequent pressure on the back wall of the eyeball and the optic nerve. Antiplatelet and anticoagulation therapy should be considered in patients with significant blood vessel disease or with presence of antiphospholipid antibodies. This treatment should be coordinated with the attending physician or cardiologist. Acetylsalicylic acid, clopidogrel, warfarin, apixaban, dabigatran, darexaban, rivaroxaban, ximelagatran, and bevacizumab are drugs that should be considered in severe vascular occlusive retinopathy [49].

Plasmapheresis and plasma exchange that have been described are also therapeutic options that may be considered in severe disease. Plasmapheresis has been used in conjunction with immunosuppressants in the treatment of patients with severe SLE-related retinal vasculitis [4,48,50]. Panphotocoagulation and pars plana vitrectomy should also be considered in severe complications of ocular ischemia. When needed, surgery should be performed to control neovascularization and vitreous hemorrhage to limit further vision loss and other possible complications.

### 2.8. Choroid

In patients with uncontrolled hypertension secondary to SLE nephropathy, and with CNS vasculitis, exudative posterior choroidal detachment may be observed. Lupus choroidopathy is commonly associated with SLE retinopathy, but its degree is reported to be mild to moderate with no significant deterioration in visual acuity.

As a complication of SLE, choroidopathy usually presents with single or multiple areas of serous detachment of the retinal pigment epithelium and neurosensory retina or retinal pigment epitheliopathy. The emerging choroidal effusion leading to anterior displacement of the lens–iris diaphragm may cause secondary angle-closure glaucoma with subsequent intraocular hypertension [51]. In 2019, an article was published that confirmed that the choroid in SLE patients appears to be thinner, particularly in the subset of patients with nephritis and taking anticoagulants, suggesting more advanced systemic vascular disease. These results probably reflect existing atrophy of choroidal tissue as well as defective vascular auto-regulatory mechanisms [52].

Clinical diagnostics in conjunction with ophthalmic imaging (FFA, indocyanine green angiography (ICG), and fundus optical coherence tomography (OCT)) are of key importance in the diagnosis of choroid and retinal pathology [53]. It is believed that the pathogenesis of choroidopathy is multifactorial. Uncontrolled hypertension, immune complexes deposited in choriocapillaries and antibodies to retinal pigment epithelium have been identified as contributing factors. It is interesting that focal areas of ICG choroidal hyperfluorescence may indicate the deposition of immune complexes in the deeper layer of the choroid stroma [54]. The presence of choroidopathy usually indicates an active inflammatory process in SLE and may herald the onset of SLE nephropathy, so requires aggressive systemic treatment and has been shown to respond to a therapeutic combination of corticosteroids and other forms of immunosuppression. The usual treatment regimen is intravenous pulses of methylprednisolone followed by oral dosing of prednisolone in combination with cyclophosphamide. In the case of asymmetric disease, the systemic corticosteroid may be supplemented by local injection of the corticosteroid into the space under the Tenon’s capsule or behind the orbital septum, depending on the preferences of the ophthalmologist. If the planned therapy lasts longer than a couple of months, it is necessary to use non-corticosteroid immunosuppressants. Appropriate immunosuppressive treatment leads, in most cases, to the resolution of lupus choroidopathy and then regaining vision [7,54,55].

Other causes of vascular lesions to be distinguished from SLE include multifocal central serous retinopathy, sympathetic inflammation, ankylosing spondylitis, reactive arthritis, Behçet’s disease, Harada’s disease, sarcoidosis, toxoplasmosis, toxocarosis and choroidal metastasis. Due to the similarity of uveal symptoms, juvenile idiopathic arthritis (JIA) should also be considered in differential diagnosis in pediatric patients. Nguyen et al. described 28 patients with lupus choroidopathy, all of whom had active systemic vascular disease and found that 64% of patients had visual acuity of 20/40 or better. Choroidopathy resolved in 82% of patients after systemic control of the underlying disease was achieved [56].

The incidence of SLE in patients with uveitis tends to range to some extent, mainly from 0.1% to 5%. There are very few reports of iritis or iritis and ciliary body inflammation secondary to SLE, especially in adults. Therefore, SLE should be considered a rare cause of uveitis. Anterior uveitis in patients with SLE is usually mild and rarely leads to a deterioration in visual acuity, and also may present as synechiae or a fibrinous inflammatory exudate in the anterior chamber of the eye. Inflammation tends to improve with systemic immunosuppressants (treatment may require NSAIDs, steroids and other immunosuppressants), but it has also been reported that atypical, refractory to treatment cases cause severe visual impairment [57].

### 2.9. Optic Nerve/CNS

Optic nerve involvement is a relatively rare manifestation of SLE and consists primarily of optic neuritis and ischemic optic neuropathy resulting in progressive loss of vision, sometimes leading to complete blindness. Other less common neurophthalmic symptoms may include abnormal pupil response, palsy of the cranial nerves, abnormal eye movement, internuclear ophthalmoplegia, blepharospasm, transient mononuclear vision loss, cortical blindness and visual field defects. Clinically, a patient with optic neuritis usually complains of unilateral, severe vision loss and eye pain that is aggravated by extreme eye movements. Color perception is also disturbed. Optic nerve involvement occurs in approximately 1% of SLE patients, and treatment rarely results in a significant improvement in visual acuity [58,59].

In a study by Lin et al. describing patients with SLE-related optic neuritis, only 50% of patients recovered to a full or near full visual acuity level. It seems to be very important to distinguish the etiology of optic neuritis—whether it is caused by the primary disease—SLE, or another endogenous or exogenous factor [59]. Optic neuritis in SLE is caused by an ischemic process that may lead to subsequent demyelination. The degree of axon loss correlates with the patient’s BCVA (Best Corrected Visual Acuity) score. Progression to optic nerve atrophy may occur in up to 50% of patients. An option to treat optic neuritis is the early administration of corticosteroids—intravenous pulsatile steroid therapy for the first days, followed by continuous oral therapy with dose reduction, with additional cyclophosphamide therapy and methotrexate [3,60].

Ischemic optic neuropathy and optic nerve chiasm in SLE have also been reported [61]. Optic neuropathy in SLE presents with sudden loss of vision due to ischemic blood flow restriction followed by hypoperfusion of the retina and optic nerve head (AION) or retrobulbar part of optic nerve (PION). In 2019, there was a case described of a 47-year-old woman with optic neuropathy presenting as relative afferent pupillary defect (RAPD) in the left eye and bilateral arcuate defects in visual field examination, an ocular harbinger of SLE [62]. Optic nerve dysfunction may be an initial symptom of systemic disease in some SLE patients, but it usually occurs during the course of the disease. Ophthalmoscopy can reveal whitening and swelling of the optic nerve with margins that are difficult to trace. Confirmation of ischemic neuropathy can be obtained by performing visual evoked potential in electrophysiological ophthalmologic studies showing reduced amplitude or increased latency of the optic nerve response. On FFA, leakage of dye around the optic discs can be observed. The basic mechanism of AION and PION is the obstruction of the small blood vessels supplying the optic nerve. The standard treatment of optic neuropathy in SLE is high-dose intravenous corticosteroids (methylprednisolone) followed by prolonged treatment with oral steroids. It is possible to relapse when the doses of steroids are reduced, which requires a combination of steroids and immunosuppressants. Studies have shown the therapeutic success of the use of immunosuppressants such as cyclophosphamide, azathioprine cyclosporine and methotrexate [63,64].

SLE can also lead to the involvement of the cranial nerves and cause double vision (diplopia). Eye movement abnormalities are common in SLE and have been observed in up to one fifth of patients as a result of ischemic disease of the microcirculation of the brainstem. Palsy of the third and sixth cranial nerve in the course of SLE is a relatively common abnormality [7,65]. Pseudotumor of the brain has been described in both children and adults with SLE and may be both the first symptom of the disease and the result of an increasing inflammatory reaction in the body [66].

Internuclear palsy can be observed in the course of SLE with the presence of diplopia, dizziness, ophthalmoplegia and ataxia. Nystagmus may also occur, but it is reported very rarely and is associated with low visual acuity of the patient. Inflammation of the blood vessels behind the optic chiasm can lead to stroke, which can result in amblyopia or hemispheric blindness. Other rare manifestations of SLE including neuro-ophthalmic symptoms reported in the literature include: Miller–Fisher syndrome, Horner’s syndrome, polyneuropathy, external ophthalmoplegia, pupillary response abnormalities, and transient monocular blindness [59,67].

### 2.10. Intraocular Infections

One of the most dangerous complications for vision in patients with SLE on immunosuppression are opportunistic intraocular infections. The most common etiology of endophthalmitis are viruses—CMV, HSV and Varicella zoster. Mycobacterium tuberculosis and Nocardia infections are less common causes in the course of SLE. Pars plana vitrectomy is the therapeutic option for acute endophthalmitis [3,7].

Manifestations of SLE in individual parts of the eye are summarized in the following Table 1.

## 3. SLE Treatments and Ophthalmic Complications

### 3.1. SLE Therapy and Ophthalmic Adverse Events (OAE)

While no targeted drug for SLE is available yet, treatment has improved significantly in recent years. The accepted goal of treating patients with SLE is to silence the immune activity in order to induce and maintain remission of the disease and prevent relapse.

Due to the different severity of disease symptoms, different therapeutic approaches are required, primarily holistic ones. In the case of ophthalmic manifestations of the disease, appropriate treatment of systemic symptoms alleviates the eye disease. The ophthalmologist is responsible for ensuring that any form of ocular involvement is treated appropriately and a determination by a rheumatologist that SLE is not active should not discourage an ophthalmologist from implementing systemic treatment, if indicated. Moreover, visual symptoms may be a harbinger of a potentially severe systemic exacerbation, requiring careful clinical monitoring. Due to the systemic nature of the disease, treatment is difficult and the cooperation of specialists in various fields (ophthalmologists, rheumatologists, nephrologists, dermatologists, etc.) is often required. Aggressive therapies that are often used in the systemic treatment of SLE are rarely used for specific isolated ocular symptoms [68,69].

The drugs used in the treatment of SLE are mainly glucocorticosteroids and hydroxychloroquine derivatives in combination with other immunosuppressants such as azathioprine, cyclophosphamide, calcineurin inhibitors, mycophenolate mofetil, methotrexate, and cyclosporin (CsA). The attending physician should pay attention primarily to the initial activity of the disease, involvement of systems and organs, and the patient’s age [69].

To monitor patients with SLE in clinical practice, treatment quality indicators have been developed to reduce the body’s adverse response to treatment [70].

Almost all of the medications used to treat SLE can have a negative effect on the eyesight, and the ophthalmic side effects of most of them are well documented in the literature. They relate to a specific anatomical structure of the eye or the entire organ of vision. It is necessary that the attending physician adequately plan screening of asymptomatic patients to recognize the early signs and symptoms of eye toxicity prior to the initiation of therapy for SLE [71]. Currently, patients with SLE are treated more often with hydroxychloroquine than with chloroquine, which significantly reduces the incidence of retinal complications [72]. However according to study of Yusuf et al., hydroxychloroquine retinal toxicity is far more common than previously considered; an overall prevalence of 7.5% was identified in patients taking HCQ for greater than 5 years, rising to almost 20% after 20 years of treatment. Toxicity was not related to age and weight of patients and daily dosage of medication [73]. Summarizing the recent literature reviews, HCQ dosing and screening guidelines still should be updated [74,75].

The most common ophthalmic complications of SLE treatment seem to be related to the use of steroid therapy, but fortunately, more often, glucocorticosteroids are replaced by biological drugs, which have less side effects on the eyesight. Possible ophthalmic side effects of drugs used in the treatment of SLE are presented in Table 2.

### 3.2. Treatment of Ophthalmic Lesions in the Course of SLE

Ocular symptoms of SLE can be a warning that the underlying inflammation in the body is insufficiently controlled and indicates that systemic treatment should be escalated. If there is no evidence of systemic inflammation, some symptoms (especially ocular surface and anterior segment disease) can be adequately treated with standard topical therapies.

In addition to the standard treatment regimens for ophthalmic lesions, there are many case reports of rituximab used in ophthalmic complications of SLE. Ocular vasculitis in SLE should be treated primarily with corticosteroids in combination with other biological agents [71].

In the literature were also described two adolescent girls with retinitis who after initial treatment with intravenous methylprednisolone, received combination therapy with rituximab, cyclophosphamide and oral corticosteroids [76].

In the case of ophthalmic complications of systemic treatment of SLE—posterior capsular cataract (Figure 4)—the treatment of choice is phacoemulsification. When glaucoma develops as a result of steroid usage, the treatment in the initial phase includes pharmacology, then surgical treatment in the absence of adequate IOP reduction to the desired level [77,78].

## 4. Future Directions

Currently, research on SLE is focused on the pathogenesis of this disease. Drugs that modulate immunity by modifying the action of T and B lymphocytes, and methods using stem and mesenchymal cells seem to be the basis of future clinical trials. Updates are also taking place in the improvement of the diagnostic criteria of the disease and the development of new therapy regimens [2,71,79,80]. Treatment of ocular pathologies in SLE relies mainly on systemic therapy.

The latest ophthalmic imaging technologies identify early changes in the morphology and vascular circulation of the retina and choroid. Microperimetry is a procedure for assessing the sensitivity of the retina during a direct fundus examination. These instruments allow comparison of the clinical fundus imaging of the patient’s eye with the sensitivity of the retina shown during the perimetry test examination.

Angio-OCT (also in version “swept source”) seems to be a very promising imaging test in ophthalmology to detect retinal and choroidal abnormalities, including drug-induced changes during SLE treatment [81,82].

## 5. Conclusions

SLE is a chronic autoimmune disease that appears to be partially due to the production of autoantibodies, which cause characteristic changes in organs of the body through the deposition of immune complexes.

In conclusion, numerous cases of ocular symptoms of systemic lupus erythematosus have been described and the ophthalmic manifestations of SLE are variable (diverse and with varying degree of severity). In addition, their presence may be a sign or marker of disease activity. In the case of choroidopathy and retinopathy, ophthalmic changes may be an unfavorable prognostic systemic risk factor with the possibility of both ocular and systemic diseases. An ophthalmologist, when detecting complications that are sight-threatening, such as retinal vasculitis, scleritis, secondary glaucoma, ulcerative keratitis or corneal perforation, should immediately extend the diagnostics to consultation of doctors of other specialties, including rheumatologists. Often the underlying cause of ophthalmic changes is an autoimmune disease such as SLE. It is also postulated that all patients with newly diagnosed SLE should undergo a thorough ophthalmological examination, which includes both basic and additional examinations. Therapy of possible ophthalmic changes should be determined individually, depending on the activity of the disease and the type and dose of medications received by the patient.

A better understanding of the pathophysiology of SLE will enable scientists to develop more effective treatment options. Treatment should aim at remission or low disease activity and, if possible, minimize the harm attributed to drug side effects. Therefore, ophthalmic screening in SLE patients should be carried out early and regularly, also with the help of new imaging techniques (S-OCT, FFA, ICG, microperimetry) to investigate the correlation between structural and functional changes in the visual organ. Close collaboration between primary care physicians, internists, immunologists, ophthalmologists and rheumatologists treating patients with SLE is essential for the successful management of complex clinical situations.

## Figures and Tables

**Figure 1 ijms-23-12264-f001:**
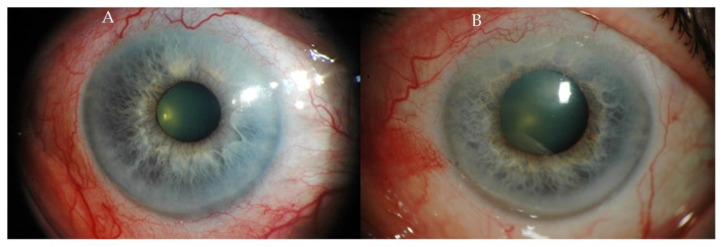
(**A**,**B**) Photographs of the anterior segment of the eye of a patient with sectoral episcleritis in the course of SLE.

**Figure 2 ijms-23-12264-f002:**
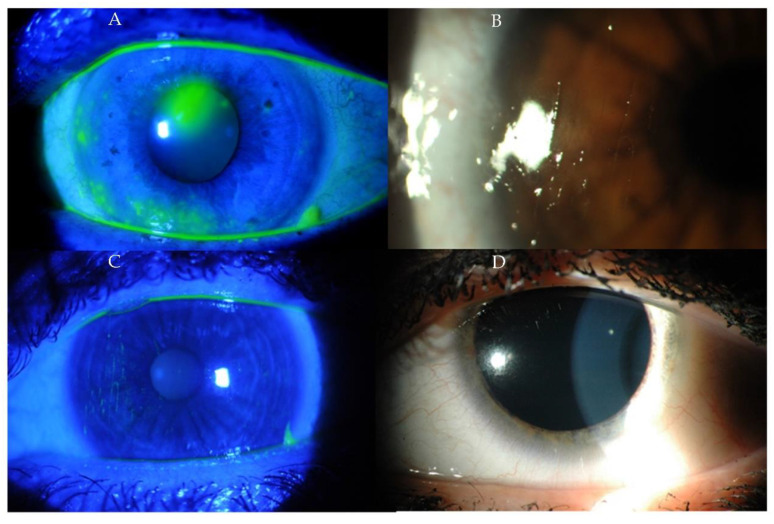
(**A**–**D**) Photographs of the anterior segment of the eye of patients with corneal filaments and mucus deposits—advanced dry eye syndrome.

**Figure 3 ijms-23-12264-f003:**
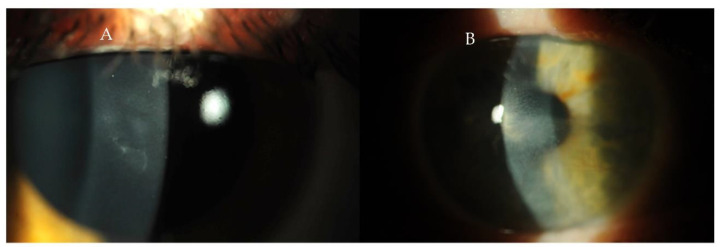
(**A**,**B**) Photographs of the anterior segment of the eye with corneal keratopathy causing irregular astigmatism in the course of SLE.

**Figure 4 ijms-23-12264-f004:**
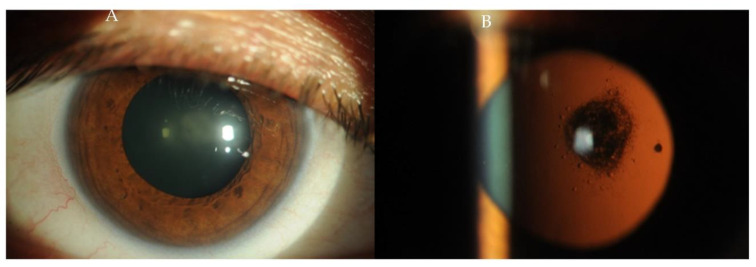
(**A**,**B**) Photographs of the anterior segment of the eye of a patient with posterior subcapsular cataract in the course of SLE.

**Table 1 ijms-23-12264-t001:** Manifestations of SLE disease in individual structures of the eye.

Structure of the Eye	Manifestations of SLE
Orbit	Subcutaneous orbital tissue inflammation, infarction of orbit muscles, myositis, panniculitis, exophthalmos, orbital edema, reduced mobility of the extraocular muscles, pseudotumor, trochleitis [7,8,9,10,11,22]
Eyelids	Heliotropic erythema, dermatitis, scaly, pigmented lesions, blepharitis, discoid rash, madarosis [18,19,20]
Cornea	Dry eye disease, corneal epitheliopathy, peripheral ulcerative keratitis, filamentous keratitis, interstitial keratitis, endothelitis [2,31,32]
Episclera and sclera	Episcleritis, scleritis, scleromalacia [2,25,30]
Retina	Occslusive retinopathy, retinal neovascularization, retinal detachment, retinal microangiopathy, Purtscher-like retinopathy [7,38,39,40,41,42]
Choroid	Iritis, iridocyclitis, choroidopathy, vasculopathy, vasculitis, choroidal effusion [2,51,55,56,57]
Optic nerve	Optic neuritis, ischemic optic neuropathy, atrophy, oedema [22,58,59,61]

**Table 2 ijms-23-12264-t002:** Types of drugs used in SLE and their possible ophthalmic side effects. Data from [68,69,70,71,72].

Types of Drugs Used in SLE Therapy	Possible Ocular Side Effects
Glucocorticoids	Subcapsular cataract, open-angle glaucoma, increased risk of ocular infection, subconjunctival hemorrhage, vitreous hemorrhage, diplopia, oculomotor palsy
Chloroquine and hydroxychloroquine	Photophobia, cornea verticillate, cataract, iridocyclitis, paralysis of oculomotor muscles, toxic maculopathy, optic neuritis
Methotrexate	Ocular pain, swelling of periorbital tissues, photophobia, conjunctivitis, decreased tear secretion, NAION, eyelid skin inflammation, blurred vision
Cyclophosphamide	Increased intraocular pressure, dry eye disease
Cyclosporin A	Deterioration of visual acuity
Mycophenolate mofetil	Keratoconjunctivitis sicca, corneal epiteliopathy
NSAIDs	Dry eye disease, corneal thinning, photophobia, pigment lesions in the macula, subconjunctival hemorrhage, hemorrhagic retinopathy, conjunctivitis
Rituximab	Conjunctivitis, swelling of periorbital tissues
Azathioprine	Infectious uveitis

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
