# Peer review of "Understanding Ocular Findings and Manifestations of Systemic Lupus Erythematosus: Update Review of the Literature"

_ijms, 2022, doi:10.3390/ijms232012264_

Round 1

Reviewer 1 Report

1/References to be improved : 

-Long paragraph without references

-Too old or not the most relevant references

2/Several phrases not really useful such as in the introduction of the paragraphs. Try to go straight to the point.

Reviewer 2 Report

Authors have summarized and updated the literature regarding ocular manifestations of systemic lupus erythematosus which is appreciable, however, there are well written reviews earlier in similar line. But since, this is an update review of literature, it is advisable to authors if they can include recent clinical case studies with regard to each ocular manifestation.

In addition, authors can also include a table or a figure to summarize various complications associated with each ocular manifestation for clear understanding.

Since authors have mentioned that various systemic drugs used for SLE causes ocular complications, a detailed subjective opinion can be mentioned based on the clinical case studies reported on its prevention and also for treating ocular complications.

It would be of additional value if authors could include images of various ocular complications as they have already mentioned about their identification.

Round 2

Reviewer 2 Report

The authors can add references to the newly added table. 
